# DAMamba: Semantic Aware One-shot Test-time Domain Adaptation for Super-resolution

## Abstract

Domain adaptation methods effectively reduce the negative impact of domain gaps on the performance of the Mamba-based super-resolution (SR) network. Considering data privacy restrictions that prevent access to source domain samples, along with the tendency of users to capture or upload only a single image for SR, we propose a semantic-aware one-shot test-time domain adaptation method for the super-resolution Mamba (DAMamba). Among them, the semantic prior-guided cross-training method employs Alpha-CLIP semantic priors with a global perspective to guide feature scanning, effectively enhancing key information extraction efficiency. Additionally, it addresses the limited diversity of target domain sample caused by single-sample constraints through pairwise patch combinations for domain adaptation training. Given the strong contextual dependency unique to the Mamba network, we propose a random blur data augmentation strategy, improving network robustness while avoiding disruptions from zero-value masking. Finally, the proposed adaptive learning strategy dynamically identifies salient and ordinary layers, further enhancing domain adaptation efficiency. Extensive experiments demonstrate the effectiveness of DAMamba, with its performance on a single target domain sample surpassing that of the state-of-the-art source-free domain adaptation methods using multiple target domain samples. Our code is available at ***.

## 1 Introduction

Super-resolution (SR) has extensive applications, such as enhancing old video clarity (Wan et al., 2022; Lin & Simo-Serra, 2024), remote sensing imaging (Xiao et al., 2024c; Miao et al., 2023a;b; Xiao et al., 2024b), and serving as a preprocessing method for downstream tasks (e.g. object detection (Zhang et al., 2023; Yang et al., 2022a; Shyam & Yoo, 2024), semantic segmentation (Kim et al., 2024; Qiu et al., 2024; Yu & Wang, 2024)).

With its input-aware property, the computational complexity of Mamba (Gu & Dao, 2023) scales linearly with the input size, making it highly suitable for long-sequence tasks (Yu & Wang, 2024). SR tasks, which reconstruct each pixel in an image, can also be considered as a long-sequence task, where a larger receptive field greatly enhances performance (Cheng et al., 2024; Guo et al., 2025). Consequently, numerous Mamba-based SR networks have emerged (Cheng et al., 2024; Guo et al., 2025; Ren et al., 2024; Lei et al., 2024; Huang et al., 2024b; Xiao et al., 2024a).

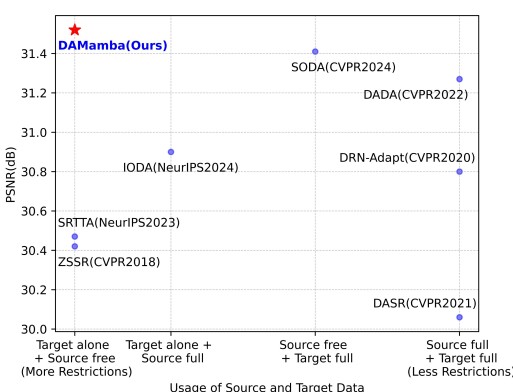

Figure 1: Performance comparison. Source domain: Olympus, target domain: Panasonic.

However, Mamba-based SR networks often face domain gap issues. Differences in imaging devices, lighting conditions, and camera shake cause discrepancies between the feature distribution of training data (source domain) and real-world (target domain), leading to significant performance drops

for SR networks. Additionally, for privacy reasons, source domain training data is typically confidential, with access prohibited. Thus, source-free domain adaptation methods (Yang et al., 2022b; Ma et al., 2024; Zhang et al., 2022b; Qu et al., 2024; Ai et al., 2024; Xia et al., 2024; Tang et al., 2024) have emerged, using target domain pseudo-labels generated by a teacher network to perform domain adaptation training without accessing source domain samples. Futhermore, in real-world scenarios, such as user-captured or uploaded images, only a few or even a single sample is typically available. Consequently, one-shot test-time domain adaptation methods (Shocher et al., 2018; Deng et al., 2023) have emerged, creating pseudo low-resolution (LR) and high-resolution (HR) images approximating the target domain distribution to adapt the network to the target domain, demonstrating significant practical application value.

CLIP (Radford et al., 2021) and its variant Alpha-CLIP (Sun et al., 2024b) were trained on millions of samples covering diverse lighting conditions, imaging devices, and extensive degradation models, endowing them with high robustness and superior feature representation capabilities. Their rich prior knowledge has been widely utilized in downstream tasks, such as object detection (Liu et al., 2024b; Pan et al., 2024), image generation (Tao et al., 2023; Chen et al., 2024; Yang et al., 2024b; Zhang et al., 2024c), semantic segmentation (Zhang et al., 2024d; Lin et al., 2023; Wang & Kang, 2024; Zhang et al., 2024a), and even domain adaptation for object detection (Li et al., 2024a; Luo et al., 2023), classification (Wu et al., 2024), semantic segmentation (Fahes et al., 2023; Yang et al., 2024c), and image generation (Gal et al., 2022; Guo et al., 2023; Yang et al., 2023). However, semantic prior-guided Mamba-based SR networks remains limited.

Therefore, we propose a semantic prior-guided cross-training method, introducing Alpha-CLIP (Sun et al., 2024b) semantic prior knowledge with a global perspective into the domain adaptation training of Mamba-based SR networks. Among them, given the linear scanning characteristic of the Mamba network, we propose the Semantic-Aware (SA) Module, which efficiently integrates global semantic priors into the scanning process, significantly enhancing the update efficiency of hidden states. Additionally, to address the limited diversity of target domain sample caused by single-sample constraints, We randomly designate sampled patches as primary and auxiliary patches, treating the auxiliary patch as background features, and increase target domain diversity by randomly combining the primary patch with varying background features. Inspired by the MAE (He et al., 2022) and considering the strong contextual dependency of the Mamba network, we propose a random blur data augmentation strategy for Mamba-based SR networks. This approach enhances network robustness while preventing disruptions from zero-value masking. Finally, inspired by (Yu et al., 2024), we propose an adaptive learning strategy that divides the network into salient and ordinary layers. By accelerating the adjustment of salient layers and preserving features of ordinary layers, the adaptive learning strategy enables efficient domain adaptation training.

The main contributions can be summarized as follows:

- We propose DAMamba to address test-time domain adaptation with a single target domain sample. To the best of our knowledge, DAMamba is the first domain adaptation method for Mamba-based SR networks and the first to integrate CLIP-related work into Mamba-based SR networks.

- (1) We propose a semantic prior-guided cross-training strategy to address the limited diversity caused by a single target domain sample, enhancing target domain feature diversity through pairwise patch combinations and using Alpha-CLIP global semantic priors to guide feature scanning in the SSM module, improving key feature extraction efficiency. (2) We introduce a random blur data augmentation strategy for Mamba, enhancing network robustness while preventing disruptions from zero-value masking. (3) We propose an adaptive learning strategy to further improve network adaptation efficiency.

- Extensive experiments demonstrate the superior performance of the proposed DAMamba. With a single target domain sample, DAMamba outperforms the state-of-the-art source-free domain adaptation methods trained with multiple target samples. Additionally, DAMamba imposes no restrictions on the pre-training of the source domain network, enabling direct domain adaptation with plug-and-play functionality.

(Refer to the Appendix A for related work on Mamba networks and domain adaptation.)

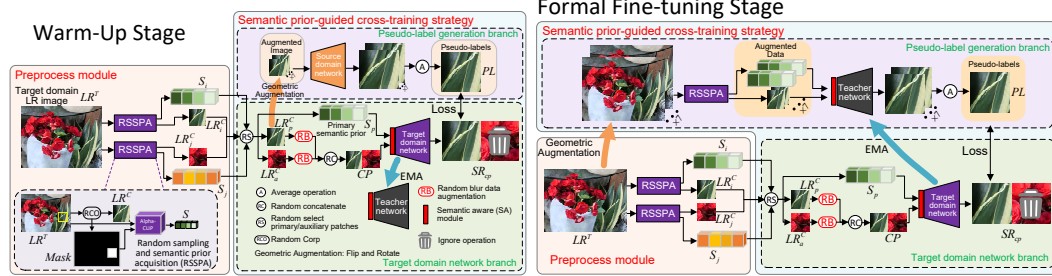

(a) Warm-up stage          (b) Formal fine-tuning stage

Figure 2: Overall structure. In this case, the $i-th$ LR patch $LR_i^C$ is designated as the primary patch $LR_p^C$, and the $j-th$ LR patch $LR_j^C$ as the auxiliary patch $LR_a^C$ for explanation. The variables in the figure correspond to Eq. 1-13. The auxiliary semantic prior information $S_a$ is omitted in the figure, as it undergoes no subsequent operations. (Please zoom-in on screen for better visualization)

## 2 METHOD

### 2.1 PROBLEM SETTING

Given a source domain dataset $D^S = \{LR_i^S, HR_i^S\}_{i=1}^{N^S}$, a target domain dataset $D^T = \{LR_i^T, HR_i^T\}_{i=1}^{N^T}$, a source domain SR network $M^S$ trained solely on the source domain dataset, a target domain network $M^T$ initialized from $M^S$, with $M^T$ additionally incorporating the proposed SA module. Among them, $N^T$ and $N^S$ represent the number of samples contained in the target domain and the source domain, respectively.

This work addresses the problem of domain adaptation training using only a single target domain LR image $LR_i^T$ (without a corresponding HR label $HR_i^T$) and without source domain $D^S$ access, aiming for network $M^T$ to outperform $M^S$ on the target domain $D^T$ after domain adaptation training.

### 2.2 OVERVIEW

Considering clarity and conciseness, we omit the pre-training process of the source domain network and focus only on the domain adaptation training process (DAMamba). The pre-training of the source domain network follows the original papers (Cheng et al., 2024; Guo et al., 2025; Ren et al., 2024). As shown in Figure 2, due to the uninitialized SA modules in the target domain and teacher networks, DAMamba consists of two stages: a warm-up stage (Section 2.3) and a formal fine-tuning stage (Section 2.4). Both stages employ a semantic prior-guided cross-training strategy for domain adaptation training. In the warm-up stage, pseudo-labels generated by the source domain network guide the SA module initialization, while the teacher model is initialized using an exponential moving average (EMA). In the formal fine-tuning stage, pseudo-labels generated by the teacher network guide the target domain network for domain adaptation training, with EMA synchronously updating the teacher network throughout the training process.

### 2.3 WARM-UP STAGE

CLIP (Radford et al., 2021) and its variant Alpha-CLIP (Sun et al., 2024b), trained on a massive dataset, exhibit powerful feature extraction and global representation capabilities. Moreover, high-level semantic priors exhibit stronger robustness to image degradation compared to low-level semantic features (Appendix D.1). Therefore, DAmamba leverages the extracted high-level semantic information to guide domain-adaptive training.

As shown in Figure 2a, the Warm-Up Stage can be divided into the preprocess module and the semantic prior-guided cross-training strategy. Specifically, the preprocess module extracts high-level semantic information, while the semantic prior-guided cross-training strategy utilizes this extracted semantic information to guide the domain-adaptive training process.

### 2.3.1 THE PREPROCESSING MODULE

The preprocessing module repeatedly inputs the target domain LR image into the Random Sampling and Semantic Prior Acquisition (RSSPA) module twice to obtain two cropped LR patches, $LR_i^C$, $LR_j^C$, along with their corresponding semantic prior information, $S_i$, $S_j$. In the RSSPA module, if the cropped patches are directly input into the image encoder of CLIP (Radford et al., 2021) to extract semantic prior information, small patch sizes reduce the accuracy of the extracted semantic priors. This is similar to the classic Chinese story 'The Blind Men and the Elephant,' where each blind man perceives the part he touches differently—the leg as a pillar, the trunk as a snake, and the tail as a rope. Therefore, we should provide CLIP with a global perspective input. However, extracting semantic prior information from the entire image makes this prior information less targeted, as each patch with different distributions corresponds to the same semantic prior information. This is unfavorable for guiding domain adaptation training.

Therefore, DAMamba introduces an enhanced version of CLIP, Alpha-CLIP (Sun et al., 2024b), which incorporates an attention mechanism (Refer to Appendix B.1 for the preliminary of Alpha-CLIP). By providing Alpha-CLIP with an additional location mask, it can focus on designated areas. As shown in Figure 2a, RSSPA randomly crops the target domain LR images to obtain LR patches and corresponding location masks. Then, RSSPA inputs the location mask and target domain LR image into the image encoder of Alpha-CLIP to focus its attention on the semantic information at the patch's location,

$$LR^C, Mask = Rco(LR^T), \tag{1}$$

$$S = Alpha\text{-}CLIP(LR^T, Mask), \tag{2}$$

$Rco$ denotes the random cropping of the target domain LR image $LR^T$ to obtain the LR patch $LR^C$ and location mask $Mask$.

### 2.3.2 SEMANTIC PRIOR-GUIDED CROSS-TRAINING STRATEGY

**(1) Motivation:** With only a single image in the target domain, limited feature diversity may lead to mode collapse, degrading network performance.

Existing SR methods typically crop an image into fixed-size patches for network training, where each patch exhibits diverse feature distributions. The proposed semantic prior-guided cross-training strategy aims to enhance target domain diversity by adaptation training on pairwise combinations of patches within a single image. Randomly designating primary and auxiliary patches, the auxiliary patch is treated as background feature, enhancing the diversity of the training sample distribution by combining different background features. Notably, since both patches originate from the target domain image without introducing features from other domains, the combined feature distribution remains closely aligned with the original target domain, avoiding distribution shifts that arise from incorporating samples from other domains.

**(2) The Semantic Prior-Guided Cross-Training Strategy** consists of two branches: the pseudo-label generation branch and the target domain network branch.

The target domain network branch randomly selects the primary and auxiliary patches $LR_p^C$, $LR_a^C$ and their corresponding primary and auxiliary semantic priors $S_p$, $S_a$ from the two input LR patches $LR_i^C$, $LR_j^C$ and their corresponding semantic priors $S_i$, $S_j$,

$$LR_a^C, LR_p^C, S_a, S_p = Random\_Sel(LR_i^C, LR_j^C, S_i, S_j)$$
$$where \quad a, p \in \{i, j\}, a \neq p, \tag{3}$$

$Random\_Sel$ refers to the random selection of the primary and auxiliary patches, along with their corresponding primary and auxiliary semantic priors.

The primary patch is fed into the pseudo-label generation branch to produce the corresponding pseudo-label $PL$. Subsequently, the primary and auxiliary patches undergo random blurring augmentation (Section 2.5), then concatenated in a random order. The concatenated patch $CP$, along with the primary semantic prior information $S_p$, is subsequently fed into the target domain network

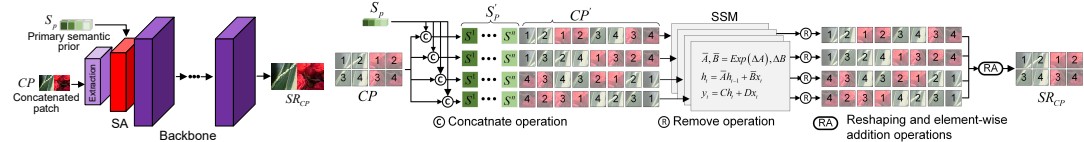

(a) Mamba network architecture    (b) SA module

Figure 3: Network architecture. The target domain network and the teacher network share the same network architecture. After the features are input into the SA module, they are unfolded in four directions (horizontal, vertical, and reverse), with a unidirectional scan performed on each unfolded feature. The variables in the figure correspond to Eq. 1 - 13.

for SR processing,

$$PL = Plgb(LR_p^C), \tag{4}$$

$$LR_a^B, LR_p^B = Random\_Blur(LR_a^C, LR_p^C), \tag{5}$$

$$CP = Random\_Cat(LR_a^B, LR_p^B), \tag{6}$$

$$SR_{cp} = M^T(CP, S_p), \tag{7}$$

$Plgb$ represents the pseudo-label generation branch, and $M^T$ represents the target domain network. $Random\_Blur$ and $Random\_Cat$ represent random blur augmentation and random-order concatenation.

The combined SR patch is then split, and the SR pixels of the primary patch region $SR_p$ are used to calculate the L1 loss against the pseudo-label. While the target domain network is updated using gradients, the EMA algorithm simultaneously updates the teacher network.

$$SR_p = Split\_and\_get\_primary\_patch(SR_{cp}), \tag{8}$$

$$Loss_{warmup} = L_1(SR_p, PL), \tag{9}$$

$Split\_and\_get\_primary$ splits the SR result of the primary patch and ignores the SR result of the auxiliary patche from the combined SR patch $SR_{cp}$.

It is worth noting that, to enhance the network's sensitivity to semantic prior information during domain adaptation training, only the primary semantic prior information is fed into the target domain network, enabling the SA module (Section 2.3.3) to filter the mixed features based on the incoming semantic prior information, thereby filtering out irrelevant background features i.e., the features of the auxiliary patch.

In the pseudo-label generation branch, since the target domain and teacher networks introduce an uninitialized SA module, the source domain network is used for pseudo-label generation. The primary patch is rotated and flipped to generate 7 augmented samples. These augmented samples undergo SR inference using the source domain network, and their aligned results are averaged to form a pseudo-label,

$$Plgb(LR_p^C) = Avg(M^S(Aug_{Geo}(LR_p^C))), \tag{10}$$

$Aug_{Geo}$ and $Avg$ represent geometric augmentation (rotated and flipped) and average operations, respectively. $M^S$ is the source domain network.

### 2.3.3 SEMANTIC AWARE (SA) MODULE

As shown in Figure 3a, after the shallow feature extraction layer, the target domain and teachers networks use the proposed SA module to guide feature scanning based on the given primary semantic prior information.

**(1) Motivation:** Since SSM scanning is unidirectional, it can only access past hidden states, meaning SSM can only use features from previously scanned image regions to infer the current pixel value. This greatly limits SSM to access global information. (Refer to Appendix B.2 for the preliminary of SSM (Mamba)) As shown in Figure 3b, to enable SSM to reference complete image information when inferring the current pixel value, and considering the linear scanning characteristic of SSM, the SA module takes a straightforward approach: appending semantic prior information before the unfolded features. This method preloads global information into SSM. In other

words, as SSM scans the unfolded image features, it has already scanned the entire semantic priors ($S_p = \{S^1, S^2, \ldots, S^n\}$), storing the semantic information of the entire image in the hidden state.

**(2) The SA module** (As shown in Figure 3b) unfolds the features of the concatenated two patches $CP \in \mathbb{R}^{B,C,H,2\times W}$ in four directions, generating $CP^{'} \in \mathbb{R}^{B,C,4,2\times H \times W}$, where $B$, $C$, $H$, and $W$ denote the batch size, number of feature channels, height, and width, respectively.

To ensure dimensional compatibility between the primary semantic prior information and the un-folded feature maps, the SA module performs dimensional expansion on the primary semantic prior $S_p \in \mathbb{R}^{B,n}$, generating $S'_p \in \mathbb{R}^{B,C,4,n}$. The expanded semantic prior is then appended to the beginning of the unfolded features ($\mathbb{R}^{B,C,4,n} + \mathbb{R}^{B,C,4,2\times H \times W} \to \mathbb{R}^{B,C,4,(n+2\times H \times W)}$) and fed into the SSM block for scanning. After scanning, the SA module removes the features corresponding to the semantic prior ($S'_p$), reshapes the four unfolded features and merges them using element-wise addition, and finally passes the merged features to the subsequent Mamba layers.

## 2.4 FORMAL FINE-TUNING STAGE

The only difference between the formal fine-tuning stage and the warm-up stage is the pseudo-label generation branch. The source domain network parameters are tailored to source domain samples and not fine-tuned for alignment with the target domain. Therefore, to enhance network adaptation to the target domain, DAMamba generates pseudo-labels using the teacher network after SA initialization. As shown in Figure 2b, the teacher network also uses semantic priors from the RSSPA module to guide pseudo-label generation. However, unlike the target domain network input, the teacher network input consists only of the primary patch and primary semantic information, without the auxiliary patch. After generating the corresponding pseudo-labels, the target domain network calculates the L1 loss, while the EMA algorithm simultaneously updates the parameters of the teacher network.

The pseudo-label acquisition in the formal fine-tuning stage is as follows (corresponding to Eq.4 and Eq.10):

$$PL = Plgb(LR^T), \tag{11}$$

$$Plgb(LR^T) = Avg(M^{Te}(LR_p^C, S_p)), \tag{12}$$

$$where \quad LR_p^C, S_p = RSSPA(Aug_{Geo}(LR^T)) \tag{13}$$

$LR_p^C$ and $S_p$ denote the primary patch and primary semantic information, respectively, while $M^{Te}$ refers to the teacher network.

## 2.5 RANDOM BLUR DATA AUGMENTATION STRATEGY

Inspired by MAE (He et al., 2022), a data aug-mentation strategy for the mamba-based SR networks is proposed. Existing mask-based augmentation methods typically set pixels to zero. However, the Mamba network updates its hidden state using past hidden states and cur-rent input, so encountering features with signif-icant pixel differences (such as zeroed features) can disrupt the hidden state (Appendix D.3). Therefore, we propose a random blur data aug-mentation strategy that blurs randomly selected areas. This approach enhances network robust-ness while preserving the general information of pixel regions, avoiding disruption from ex-treme pixel values.

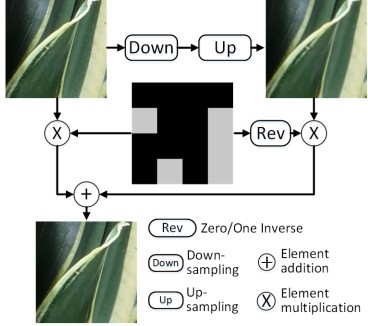

Figure 4: Random blur data augmentation strategy

As shown in Figure 4, the random blur data augmentation method first performs $2\times$ down-sampling and up-sampling on the input LR patch $LR^C$ to create a paired blurred patch. A random grid $Grid$ of the same size as the LR patch is generated, with a random blur augmentation ratio $Ratio^B$. Each grid cell has a probability $Ratio^B$ of being set to 0 and $(1 - Ratio^B)$ of being set to 1. This grid is

**Input:** Target domain network $M^T$; The number of layers in the target domain network $N^{Layer}$; The ratio of salient layers $ratio^{sa}$; The increase and decrease ratios of the learning rate, $ratio_{increase}$ and $ratio_{decrease}$;

**Output:** Network parameters with adjusted learning rates $para$;

1 **for** $i \in [1, N^{Layer}]$ **do**

2    Calculate the mean value of the parameters in the $i - th$ layer of the target domain network $Mean^i$;

3 **end**

4 $Salient\_layer\_name = Get\_TOP(Mean, ratio^{sa})$;

   // $Mean = \{Mean^1, Mean^2, \cdots, Mean^{N^{Layer}}\}$

5 **for** $name, para \in M^T.name\_and\_parameter()$ **do**

6    **if** $name \in Salient\_layer\_name$ **then**

7      $para.Lr = para.Lr \times ratio_{increase}$;

8    **else**

9      $para.Lr = para.Lr \div ratio_{decrease}$;

10    **end**

11 **end**

12 **return** $para$;

**Algorithm 1:** Adaptive learning strategy

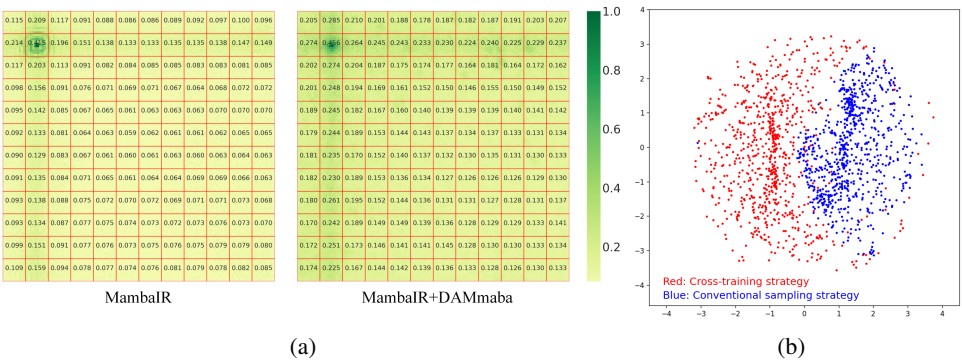

MambaIR      MambaIR+DAMmaba

Red: Cross-training strategy
Blue: Conventional sampling strategy

(a)            (b)

Figure 5: **(a)** The Effective Receptive Field visualization (Luo et al., 2016). A pixel in the upper-left corner of the image is selected as the observation pixel. The dark area shows the receptive field of the observation pixel, with higher intensity (darker brightness) indicating a greater influence on the observation pixel. For clarity, the image is divided into equal-sized grids, displaying the average intensity value in each grid. (Please zoom-in on screen). **(b)** Sample distribution. We collected 600 samples using cross-training and conventional sampling strategies, and performed dimensionality reduction visualization with T-SNE (Van der Maaten & Hinton, 2008). Notably, a more dispersed scatter plot indicates greater diversity in the collected samples. (The cross-training strategy refers to randomly concatenating two patches and inputting them into the SR network for training.)

then used to select values from the original and blurred LR patches to produce an augmented patch. The process of random blur data augmentation is as follows:

$$LR^B = Grid \times LR^C + Reverse(Grid) \times Up(Down(LR^C)), \tag{14}$$

$Reverse$ denotes the 0-1 inversion operation, while $Up$ and $Down$ represent up-sampling and down-sampling operations, respectively. $LR^B$ is the augmented image.

## 2.6 ADAPTIVE LEARNING STRATEGY

PEFT (Zhang et al., 2024e; Peng et al., 2024) fine-tunes a part of parameters to achieve target domain adaptation while preserving feature processing capabilities from pre-training. Global fine-tuning methods adjust all network parameters and use regularization terms to limit parameter variation, thereby preserving the feature-processing capability. However, Yu et al. (2024) suggests that PEFT may underfit due to limited parameter adjustment, while global fine-tuning requires careful manual regularization design.

The adaptive learning strategy divides the network into salient and ordinary layers. The salient layers play a more crucial role in the network's inference, as their high parameter values greatly influence the propagation of input features across layers. Therefore, the adaptive learning strategy aims to adjust the learning rate to enhance the fitting of salient layers to the target domain during domain

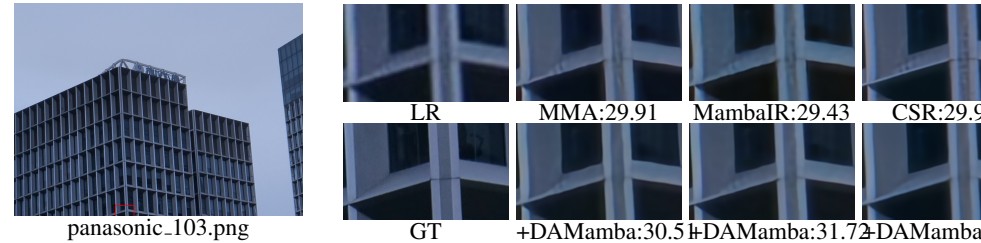

Figure 6: Visual comparison of DAMamba. The large image on the left is the LR image, and the sub-images on the right are LR, MMA (Cheng et al., 2024), MambaIR (Guo et al., 2025), CSR (Ren et al., 2024)(first row), GT, MMA + DAMamba, MambaIR + DAMamba, CSR + DAMamba (second row). The value following the name represents the PSNR metric of the current patch.

Table 1: Effectiveness validation of DAMamba. Source domain: Olympus, target domain: Panasonic, network architecture: MambaIR (Guo et al., 2025). We conducted three repeated experiments and reported the mean and standard deviation.

| Method | PSNR | SSIM |
|---|---|---|
| Baseline | 30.80 | 0.8585 |
| Add SPCT | 31.20 | 0.8599 |
| Add Random blur | 31.24 | 0.8597 |
| Add Adaptive learning | 31.39 | 0.8611 |
| Add Pre-training | 31.52±2.6e-3 | 0.8642±5.4e-4 |

Table 2: Ablation experiments on semantic-guided cross-training strategy.

| SP | CT | PSNR | SSIM |
|---|---|---|---|
| | | 30.80 | 0.8585 |
| ✓ | | 31.12 | 0.8597 |
| | ✓ | 31.17 | 0.8617 |
| ✓ | ✓ | 31.20 | 0.8599 |

adaptation training, while minimizing parameter changes in ordinary layers to avoid overfitting. As shown in Algorithm 1, the adaptive learning strategy first computes the average value of the parameters for each layer, then ranks these average values. Layers with higher average values are selected as salient layers, with their learning rates increased, while the learning rates of other layers are reduced.

It is noteworthy that the adaptive learning strategy is applied before both the warm-up stage and the formal fine-tuning stage to dynamically adjust the learning rate of the target domain network. (Refer to Appendix C for the pseudocode of the overall training pipline)

## 3 EXPERIMENTS

### 3.1 EXPERIMENT DETAILS

We evaluate our method on the DRealSR (Wei et al., 2020), Set5 (Bevilacqua et al., 2012), Set14 (Zeyde et al., 2010), B100 (Martin et al., 2001), Urban (Huang et al., 2015), Manga109 (Matsui et al., 2017), and DIV2K (Timofte et al., 2017) datasets. DRealSR is a large-scale real-world SR benchmark, collected using various DSLR cameras (Panasonic, Sony and Olympus et al.) in real-world scenarios. In the ablation experiments, one data branch serves as the source domain, while the remaining data branches constitute the target domains. Notably, source domain samples are used only for pre-training the source domain network and are inaccessible during domain adaptation training. Only a single LR image from the target domain is used for adaptation training, with the HR image used solely for performance testing. After performing domain adaptation training on all target domain images, their average accuracy is taken as the final result.

### 3.2 ABLATION EXPERIMENT

#### 3.2.1 ABLATION EXPERIMENTS FOR EACH MODULE OF DAMAMBA

To validate the effectiveness of the proposed modules, ablation experiments were conducted for each module. The performance of the source domain network $M^S$ on the target domain without domain adaptation training serves as the baseline. As shown in Table 1, incorporating the semantic prior-guided cross-training (SPCT) strategy significantly improves network performance, with a 0.4 dB increase in PSNR. With the addition of random blur data augmentation, network robustness further improves, yielding a 0.44 dB increase in PSNR over baseline performance. With the

Table 3: Performance comparison on the Olympus branch. ST: Single target domain sample, SF: Source-free. The best and second best performance are in red and blue colors, respectively.

| Method | ST | SF | PSNR | SSIM | LPIPS |
|--------|----|----|------|------|-------|
| CinCGAN (Yuan et al., 2018) | ✗ | ✗ | 29.37 | 0.799 | 0.381 |
| DRN-Adapt (Guo et al., 2020) | ✗ | ✗ | 30.80 | 0.822 | 0.356 |
| DADA (Xu et al., 2022) | ✗ | ✗ | 3 1.27 | 0.824 | 0.348 |
| Mair (Li et al., 2025) | ✗ | ✗ | 31.05 | 0.8626 | 0.349 |
| DATM (Huang et al., 2024a) | ✗ | ✗ | 31.23 | 0.8615 | 0.345 |
| SODA (Ai et al., 2024) | ✗ | ✓ | 31.41 | 0.832 | 0.344 |
| IODA (Tang & Yang, 2024) | ✓ | ✗ | 30.90 | 0.859 | 0.369 |
| ZSSR (Shocher et al., 2018) | ✓ | ✓ | 30.42 | 0.843 | 0.412 |
| SRTTA (Deng et al., 2023) | ✓ | ✓ | 30.47 | 0.845 | 0.456 |
| DAMamba | ✓ | ✓ | 31.52 | 0.864 | 0.341 |

proposed adaptive learning strategy incorporated into domain adaptation training, dynamic selection and optimization of salient and ordinary layers significantly enhance performance, improving PSNR by 0.59 dB over the baseline. Additionally, with pre-trained weights in the source domain network training (Add Pre-training), the PSNR reached 31.52, surpassing the performance of the state-of-the-art source-free domain adaptation method using multiple target domain samples. As shown in Figure 6, after performing domain adaptation training using the proposed DAMamba method on the target domain network (Cheng et al., 2024; Guo et al., 2025; Ren et al., 2024), texture details are noticeably improved.

### 3.2.2 ABLATION EXPERIMENTS ON SEMANTIC PRIOR-GUIDED CROSS-TRAINING STRATEGY

As shown in Table 2, we validated the effectiveness of each module in the semantic prior-guided cross-training strategy. With the introduction of semantic prior (SP) information, i.e., adding the SA module, the PSNR metric increased by 0.32 dB. As shown in Figure 5a, introducing semantic information equips the SSM with a global perspective, enhancing the impact of the entire image on the inference of a single pixel. Even the influence of distant lower-right pixels on the inference of the upper-left observation pixel is enhanced. Additionally, the cross-training (CT) strategy, which randomly combines target domain patches, further enhanced the diversity of training sample distributions (Figure 5b) and improved network performance, reaching a PSNR of 31.20 dB.

### 3.3 COMPARATIVE EXPERIMENT

DAMamba was compared with other domain adaptation methods. These include SR domain adaptation methods for general scenarios (CinCGAN (Yuan et al., 2018), DRN-Adapt (Guo et al., 2020), DADA (Xu et al., 2022)), which use both source and multiple target domain samples for domain adaptation training, the state-of-the-art source-free domain adaptation method SODA (Ai et al., 2024), which does not require source samples for adaptation training, and IODA(Tang & Yang, 2024), which only requires a single target sample. Finally, DAMamba is compared with similarly constrained methods (ZSSR (Shocher et al., 2018), SRTTA (Deng et al., 2023)) that perform domain adaptation without accessing the source domain and using only a single target sample. As shown in Table 3, DAMamba achieves satisfactory performance, surpassing the multi-sample state-of-the-art source-free domain adaptation method (SODA). (Refer to Appendix E for visualization results)

## 4 CONCLUSION

In this paper, we propose a semantic-aware one-shot test-time domain adaptation method for super-resolution Mamba (DAMamba), achieving efficient domain adaptation training without source samples and using only a single target domain sample. Our semantic prior-guided cross-training method enhances target domain sample diversity through pairwise patch combination and leverages Alpha-CLIP semantic priors with a global perspective to guide SSM feature scanning, improving key feature extraction efficiency. Additionally, the proposed random blur data augmentation strategy prevents hidden state-space disruptions caused by zero-value masks, enhancing network robustness. The adaptive learning strategy dynamically selects salient and ordinary layers, preserving feature processing capability while effectively fitting the target domain. In the future, we will implement DAMamba across different fields, including object detection and semantic segmentation, to enhance its applicability in various scenarios.

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

# A RELATED WORK

## A.1 MAMBA

Mamba was a data-aware State Space Model (SSM) (Gu et al.) that dynamically adjusted parameters based on input features and selectively updated the hidden states. It achieved efficient receptive field expansion by retaining past key information in the hidden states. Additionally, Yu & Wang (2024) demonstrated that Mamba was more computationally efficient than Transformers in long sequence and regression tasks. Therefore, numerous Mamba networks for computer vision emerged.

### A.1.1 MAMBA FOR OTHER TASKS

The Mamba model for sequence tasks employed linear sampling, i.e., single-direction sequential scanning, primarily targeting one-dimensional features like time series. Liu et al. (2024c); Zhu et al. (2024) argued that this approach was unsuitable for two-dimensional features such as images, and thus proposed bi-directional and four-directional scanning strategies. Ren et al. (2024) proposed a windowed scanning strategy, which divided the image into multiple small windows, scanning each window sequentially before moving to the next. Hu et al. (2024) argued that previous scanning algorithms lacked spatial continuity. They proposed a zigzag scanning algorithm, which preserved spatial continuity while reducing scanning redundancy by performing only single-direction scans each time. Shi et al. (2024) proposed a six-directional scanning strategy, where features were first scanned using the traditional four-directional scan methods, followed by forward and backward scans along the channel dimension. Li et al. (2024b) was the first to incorporate Fourier transform into the Mamba network. In the frequency spectrum after the Fourier transform, the central region contained low-frequency information, while the surrounding areas contained high-frequency information. The existing four-directional scanning algorithm inevitably disrupt this information correlation. They proposed a diagonal-like scanning strategy that first scanned high-frequency information, then low-frequency information, and finally returned to the high-frequency region. Xiao et al. (2024d) argued that existing manually designed multi-directional scanning introduced spatial discontinuities between adjacent pixels. Therefore, they designed an input-aware scanning order selection strategy that utilized a Vontractive Boruvka tree structure, enabling the network to autonomously select scanning positions and greatly preserving the spatial characteristics.

### A.1.2 MAMBA FOR LOW LEVEL TASKS

Guo et al. (2025) first introduced Mamba into SR tasks. They addressed the issue of scattered adjacent pixel scanning in existing algorithms by utilizing a convolutional network to aggregate neighboring features before feature scanning. Additionally, the authors identified significant redundancy in the features generated by the existing four-directional scanning algorithm and introduced a channel attention mechanism, effectively reducing feature redundancy. Subsequently, Cheng et al. (2024) further enhanced the network's performance by incorporating a pre-training strategy and a bidirectional scanning strategy. Lei et al. (2024) utilized super-resolution large model distillation to transfer features to a smaller model, achieving model light weighting. Huang et al. (2024c) first applied convolutions with different kernel sizes to the input features to extract features with various receptive field sizes. They then used wavelet transform to extract frequency features from the small kernel features, and concatenated these with multi-scale features before feeding them into the Mamba block, effectively enhancing its ability to restore texture details. Following the inspiration from Fouriermamba (Li et al., 2024b), numerous studies applied frequency information to optimize networks (Xiao et al., 2024a; Zou et al., 2024; Zhang et al., 2024b; Yamashita & Ikehara, 2024; Sun et al., 2024a). Zou et al. (2024) first validated the dominant role of low-frequency features in image restoration tasks by cross-fusing high- and low-frequency features from two images. They then applied wavelet transform to separate these features and used a complementary strategy between high- and low-frequency features during the subsequent upsampling process, effectively improving image restoration performance. The values of each pixel in the frequency spectrum generated by the fourier transform were influenced by the entire image. Additionally, Zhang et al. (2024b) noted that the generated frequency spectrum had position-invariant properties. Based on these two characteristics, they incorporated frequency features into RGB features. Similarly approached from the frequency domain, Sun et al. (2024a) demonstrated the dominant role of low-frequency information in rainy images by exchanging high-frequency and low-frequency information between rainy and

normal images. They also introduced FFT for feature processing in the frequency domain. Ji et al. (2024a) was the first to apply Mamba to super-resolution tasks in medical imaging, effectively reducing network computational cost while enhancing performance. Ji et al. (2024b) proposed a data augmentation strategy for medical image super-resolution, involving random block selection and arbitrary-intensity brightness enhancement. They also introduced an attention mechanism to fuse features generated from different sampling directions. Di et al. (2024) was the first to apply Mamba to BurstSR tasks. They argued that the alignment and information filtering between auxiliary frames and the base frame in existing networks were often done pairwise, lacking a global view. Therefore, they proposed QSSM, using the base frame as a Query to control the influence of auxiliary frames on the current state. Lu et al. (2024) was the first to apply Mamba to super-resolution tasks for light field images. Since light field images have an additional dimension compared to regular RGB images, they proposed a cross-feature extraction strategy. During feature extraction, they merged the features from the other two dimensions and performed cross-dimensional feature extraction, effectively addressing high-dimensional issues. Additionally, they split the features into four parts along the channel dimension and scanned each part in different directions, significantly improving network computational efficiency.

## A.2 DOMAIN ADAPTATION

In real-world scenarios, there were inevitable differences between the distributions of the training set (source domain) and the test set (target domain), a phenomenon known as domain shift. This discrepancy leads to significant performance degradation for networks that perform well in the source domain when applied to the target domain. Therefore, domain adaptation methods were developed.

### A.2.1 DOMAIN ADAPTATION FOR OTHER TASKS

Fine-tuning the entire network required significant computational resources, while fine-tuning parts of the network often led to insufficient fitting. Therefore, Park et al. (2024) proposed a dynamic fine-tuning selection strategy, which dynamically chose between full and partial fine-tuning based on the prediction difference between the student and teacher networks, achieving higher fine-tuning efficiency. Additionally, they introduced masked training, enhancing the robustness of the network by applying mask augmentation to the input images. Yu et al. (2024) proposed a domain-specific layer selection and update strategy, which selected domain-specific layers by calculating the similarity of output features before and after adding noise. By exclusively training on domain-specific features, significantly improved the efficiency of the domain adaptation training. Gao et al. (2024) studied domain adaptation in open test settings, where the test set included categories and style features not present in the source domain. They distinguished between known and unknown classes based on their bimodal nature and applied a training strategy using minimum entropy for known classes and maximum entropy for unknown classes. Zhang et al. (2022a) argued that network prediction boundaries should lie in sparse regions and that predictions should have high confidence. Therefore, they applied multi-class augmentation to target domain images and used entropy constraints on the predictions to maximize result consistency. Ma (2024) argued that complex data augmentation strategies, such as contrast and brightness adjustments, altered the original data distribution. Therefore, they proposed simpler augmentations like image cropping and horizontal flipping. Since data augmentation inevitably changed some feature distributions, they introduced Non-Maximum Suppression (NMS) for pseudo-label generation. Additionally, the authors used neighboring features to assist in the pseudo-label generation strategy, further improving label accuracy. Gandelsman et al. (2022) introduced the concept of auxiliary tasks during network pre-training, incorporating a method similar to MAE (He et al., 2022). Random masks were applied to target domain images, and the auxiliary reconstruction task was used to fine-tune the backbone network for better adaptation. Liu et al. (2024a) addressed the issue of error accumulation in continual test-time adaptation by proposing a masking strategy to selectively mask domain-specific features, effectively preventing error accumulation. Xia et al. (2024) argued that incorrect network predictions were often caused by the failure to properly focused on the target area. Therefore, they proposed a confidence-based loss weighting, which effectively expanded the attention area of the network to cover the target region. Khurana et al. (2021) studied domain adaptation for single target domain sample. Previous test-time domain adaptation methods primarily adapted the mean and variance of the Normalization layer to the target domain (Schneider et al., 2020), but this caused bias in single-sample cases. To

address this, the authors introduced a weighting coefficient to balance the source and target domain distributions, effectively reducing bias caused by single samples.

### A.2.2 DOMAIN ADAPTATION FOR SUPER-RESOLUTION

Shocher et al. (2018) performed bilinear downsampling on the target domain LR image to generate paired pseudo-LR images. Supervised training was then conducted using the paired pseudo-LR and pseudo-HR images to fit the target domain features. Deng et al. (2023) argued that the ZSSR network, which only included bilinear downsampling, could not generalize to real-world scenarios. Consequently, they proposed a degradation model that included BlurJPEG, NoiseJPEG, and GaussianBlur. Additionally, they introduced a degradation recognition network to identify the degradation model experienced by the target domain LR images. Based on the recognized degradation model, paired pseudo-LR images were generated, and supervised training was conducted using the target domain LR images to achieve domain adaptation. Wang et al. (2021) utilized adversarial networks to align the feature distributions of the source and target domains, enabling the network to adaptively train to the target domain. Wei et al. (2021) introduced the concept of domain distance into the domain adaptation process, using the confidence values output by the discriminator network as a similarity metric between input samples and target domain distributions. Higher similarity samples were assigned greater loss weights, effectively enhancing the adaptation performance of the network. Xu et al. (2022) introduced dual adversarial networks: Inter-domain Adversarial Adaptation (InterAA) and Intra-domain Adversarial Adaptation (IntraAA), which effectively improved the fitting efficiency of the network to the target domain. Ai et al. (2024) replaced the Softmax layer with a Cumbel Softmax layer, effectively enhancing label robustness. Additionally, they introduced wavelet transformation into data augmentation, merging features from multiple instances in the low-frequency space to further improve the robustness of pseudo-labels. Rad et al. (2021) applied the concept of reference SR by utilizing the activation layers in the classification network to select samples, forming an activation dataset that served as a sample pool during fine-tuning, effectively enhancing the perceptual quality of the generated images.

### A.2.3 DOMAIN GENERALIZATION FOR MAMBA

With the rise of Mamba, domain-related methods based on Mamba have emerged. Yang et al. (2024a) proposed the MSD module, which masked noise during the 3D point cloud scanning process by allowing the network to autonomously learn the mask, effectively reducing noise impact on network performance. Additionally, the authors introduced a dual-layer scanning strategy: the first layer scanned the current domain before proceeding to the other domain, while the second layer involved pixel-level dual-domain cross-scanning. Furthermore, to address the neglect of domain-invariant features in existing domain generalization methods, they proposed SCFA, which merged features of the same category from source and target domains, effectively alleviating the problem of overfitting to a single domain. Long et al. (2024) believed that domain variation was primarily caused by environmental changes, which had a minimal impact on target entities. Therefore, they used attribution algorithms to identify entity and environmental regions. First, they randomly disturbed the patches of the environmental region, and then randomly exchanged the environmental patches between the source and target domains. These two strategies effectively enhanced the robustness of the network against domain shift.

The above two methods relied on dual-domain features from the source and target domains during training and required a substantial number of target domain samples and their corresponding ground true labels for the identification of features belonging to the same class, which was difficult to achieve in scenarios where only target domain samples were available. Therefore, we proposed the DAMamba method to achieve domain adaptation training without requiring source domain samples and with only a single target domain sample.

## B PRELIMINARIES

### B.1 ALPHA-CLIP

As shown in Figure 7, the Alpha-CLIP network (Sun et al., 2024b) builds upon the CLIP network (Radford et al., 2021) by additionally introducing mask branch (Alpha Conv), which runs parallel

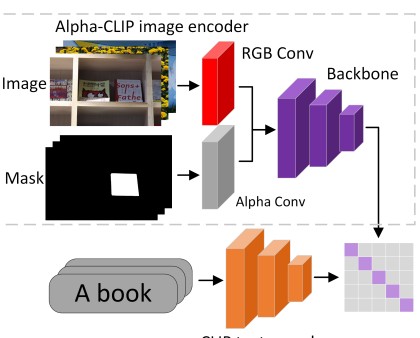

Figure 7: Alpha-CLIP

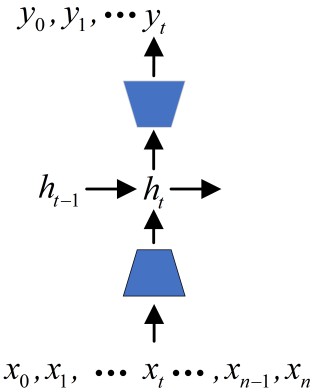

Figure 8: The inference process of Mamba. $h_t$ and $y_t$ represent the hidden state and predicted result at time step $t$, respectively.

to the original image feature extraction branch (RGB Conv) and shares a common feature backbone. The only difference is that the Alpha Conv has one input channel, while the RGB Conv has three input channels. Alpha-CLIP is trained using contrastive learning methods, with the category corresponding to the mask position treated as the ground true label. This training strategy enables Alpha-CLIP to focus more on the semantic features of the targets at the mask locations.

### B.2 MAMBA (SSM)

As shown in Figure 8, the concept of Mamba (Gu et al.) is analogous to that of RNN (Medsker et al., 2001) and was initially proposed for one-dimensional feature processing. The primary distinction between Mamba and its predecessor is the introduction of input-aware network parameters.

Mamba selectively retains the current input using input-aware parameters and updates the previous hidden state with the retained features, subsequently predicting the output value for the current time step using the updated hidden state. By selectively retaining key information at each time step, it stores global information of the previously scanned area with a smaller hidden state size. The predicted result of Mamba at time $t$ can be expressed by the following formula:

$$B = Linear_B(x_t), \tag{15}$$

$$C = Linear_C(x_t), \tag{16}$$

$$\Delta = Softplus(Par + Linear_\Delta(x_t)), \tag{17}$$

$$\overline{A}, \overline{B} = Discretize(\Delta, A, B), \tag{18}$$

$$h_t = \overline{A}h_{t-1} + \overline{B}x_t, \tag{19}$$

$$y_t = Ch_{t-1} + Dx_t, \tag{20}$$

$Linear_B$, $Linear_C$ and $Linear_\Delta$ represent three linear layers, where $\Delta$ indicates the time scale parameter, and $Discretize$ denotes the discretization operation (Zero-order hold (ZOH) method). $A$ and $Par$ represent trainable parameter matrices. Mamba uses the discretized parameter $\overline{B}$ for selective retention of the current input $x_t$ and employs $\overline{A}$ to control the retention of the previous hidden state $h_{t-1}$, generating the current hidden state $h_t$. Finally, parameters $C$ and $D$ balance the influences of the hidden state and the current input, outputting the predicted value at the current time step $y_t$.

It is worth noting that, in one-dimensional sequence tasks, when the network infers the input at time step $t$, it can only observe features between time steps 0 and $t$, while features after time step $t$ remain invisible.

Although existing methods introduce multi-directional scanning strategies into Mamba-based computer vision tasks, which consist of multiple unidirectional scanning operations, aiming to achieve global perception by summing the features obtained from each scan. Multi-directional scanning strategies typically rely on unidirectional feature scans in four directions—horizontal, vertical, and reverse—which inevitably leads to the separation of adjacent pixels after scanning, such as diagonally adjacent pixels. During unidirectional scanning, feature scans in each direction are independent, relying solely on the features previously scanned in that direction. Obtaining complete image features requires scanning the entire image. Additionally, if features without a global perspective are combined through simple addition, the resulting features still lack a global view. Furthermore, in the process where SSM selectively retains key features, selecting and retaining key features based on partial features would be extremely challenging.

Therefore, we propose a semantic information-guided cross-training strategy, which introduces the global semantic information of Alpha-CLIP. By pre-injecting this global information into the hidden state, SSM can selectively retain key features with a global perspective. This also equips each unidirectional scan with global awareness, enhancing the global perspective and representational capacity of the merged features.

## C  ALGORITHM OF OVERALL TRAINING PIPLINE

The overall training pipline of the DAMamba method is shown in Algorithm 2.

## D  ADDITIONAL EXPERIMENTS

We conducted ablation experiments on prior information acquisition methods (CLIP or Alpha-CLIP, D.1), different SA insertion layers (D.2), random blurring data augmentation (D.3), varying blurring ratios (D.4), different saliency layer ratios (D.5), training strategies (EMA and warm-up stage, D.6), as well as different network architectures (D.7), data distributions (D.8) and training/inference efficiency (D.9).

### D.1  ABLATION EXPERIMENTS ON SEMANTIC PRIOR GENERATION METHODS

we assessed the impact of semantic prior generation methods based on Self-attention, CLIP and Alpha-CLIP on domain adaptation training efficiency. As shown in Table 4, replacing Alpha-CLIP with global attention modules leads to significant performance degradation, which demonstrates the effectiveness of Alpha-CLIP's semantic features. Alpha-CLIP's comprehensive pretraining dataset, encompassing diverse object categories and degradation patterns, endows its features with both richer semantic priors and superior robustness compared to conventional global attention modules. Inputting patches (CLIP_Patch) into CLIP for semantic extraction provides a limited view due to the smaller patch size, resulting in ambiguous semantics. When using the entire image (CLIP_Full), the extracted semantic information lacks specificity, applying the same prior to all patches and thus reducing training efficiency. With Alpha-CLIP, not only is the global receptive field retained, but attention mechanisms also ensure that the extracted semantic information is specific to each patch.

**Input:** Source domain network $M^S$; Target domain network $M^T$; Teacher network $M^{Te}$; Target domain LR image $LR^T$; The number of layers in the target domain network $N^{Layer}$; The ratio of salient layers $ratio^{sa}$; The increase and decrease ratios of the learning rate, $ratio_{increase}, ratio_{decrease}$;

**Output:** Adjusted target domain network $M^T$;

```
// Adaptive learning strategy for Warm-up stage
```

1   $M^T = Adaptive\_learning\_strategy(M^T, N^{Layer}, ratio^{sa}, ratio_{increase}, ratio_{decrease})$;

```
// Warm-up stage
```

2   **for** *each step* $\in [1, Num_{Warmup}]$ **do**

3    $LR_i^C, S_i = RSSPA(LR^T)$; `// Eq.1-2 in the main paper.`

4    $LR_j^C, S_j = RSSPA(LR^T)$;

5    $LR_a^C, LR_p^C, S_a, S_p = Random\_Sel(LR_i^C, LR_j^C, S_i, S_j) \quad where \quad a, p \in \{i, j\}, a \neq p$

6    $PL = Avg(M^S(Aug_{Geo}(LR_p^C)))$; `// The only difference from the formal fine-tuning stage. Corresponding to Eq.10 in the main paper.`

7    $LR_a^B, LR_p^B = Random\_Blur(LR_a^C, LR_p^C)$; `// Random blur data augmentation strategy`

8    $CP = Random\_Cat(LR_a^B, LR_p^B)$;

9    $SR_{cp} = M^T(CP, S_p)$;

10    $SR_p = Split\_and\_get\_primary\_patch(SR_{cp})$;

11    $Loss_{warmup} = L1(PL, SR_p)$;

12    $Loss_{warmup}.backword()$;

13    $M^{Te} = EMA(M^T)$;

14 **end**

```
// Adaptive learning strategy for formal fine-tuning stage
```

15   $M^T = Adaptive\_learning\_strategy(M^T, N^{Layer}, ratio^{sa}, ratio_{increase}, ratio_{decrease})$;

```
// Formal fine-tuning stage
```

16   **for** *each step* $\in [1, Num_{Formal}]$ **do**

17    $LR_i^C, S_i = RSSPA(LR^T)$; `// Eq.1-2 in the main paper.`

18    $LR_j^C, S_j = RSSPA(LR^T)$;

19    $LR_a^C, LR_p^C, S_a, S_p = Random\_Sel(LR_i^C, LR_j^C, S_i, S_j) \quad where \quad a, p \in \{i, j\}, a \neq p$

20    $PL = Avg(M^{Te}(RSSPA(Aug_{Geo}(LR^T))))$; `// The only difference from the warm-up stage. Corresponding to Eq.11-13 in the main paper.`

21    $LR_a^B, LR_p^B = Random\_Blur(LR_a^C, LR_p^C)$;

22    $CP = Random\_Cat(LR_a^B, LR_p^B)$;

23    $SR_{cp} = M^T(CP, S_p)$;

24    $SR_p = Split\_and\_get\_primary\_patch(SR_{cp})$;

25    $Loss_{formal\_fine\_tuning} = L1(PL, SR_p)$;

26    $Loss_{formal\_fine\_tuning}.backword()$;

27    $M^{Te} = EMA(M^T)$;

28 **end**

29 **return** $M^T$;

**Algorithm 2:** Overall training pipline of the DAMamba method. All variables correspond to Eq.1-12 in the main paper.

Table 4: Ablation experiments on semantic prior generation methods.

| Method | PSNR | SSIM |
|---|---|---|
| Baseline | 30.80 | 0.8585 |
| Self-attention | 30.98 | 0.8622 |
| CLIP_Patch | 31.42 | **0.8653** |
| CLIP_Full | 31.43 | 0.8648 |
| Alpha-CLIP | **31.52** | 0.8642 |

## D.2 ABLATION EXPERIMENTS ON DIFFERENT SA INSERTION LAYERS

We conducted ablation experiments to assess the impact of adding the SA module at various layers in the Mamba network on domain adaptation performance. As shown in Table 5, adding the SA module after the initial convolution layer (Before) yields the best domain adaptation performance. Adding the SA module in intermediate layers may disrupt feature processing capabilities developed during pre-training, negatively affecting adaptation performance. Additionally, adding the SA module after the backbone network similarly avoids disrupting the feature extraction capability of the backbone (After), leading to suboptimal results.

Table 5: Ablation experiments on different SA insertion layers

| Layer | PSNR | SSIM |
|---|---|---|
| Before | **31.52** | 0.8642 |
| 1 | 31.01 | 0.8587 |
| 2 | 31.00 | 0.8581 |
| 3 | 31.02 | 0.8597 |
| 4 | 30.98 | 0.8593 |
| 5 | 31.05 | 0.8602 |
| After | 31.50 | **0.8654** |

## D.3 ABLATION EXPERIMENTS ON RANDOM BLUR DATA AUGMENTATION

As shown in Table 6, we conducted ablation experiments on the random blur data augmentation strategy. First, we applied a data augmentation method similar to MAE (Opaque masking), randomly covering parts of the image and setting masked pixel values to zero. Compared to the proposed random blurring method, the zero-value masking operation caused a PSNR drop of 0.51 dB. Due to strong contextual dependency characteristic of Mamba, the zero-value outliers significantly disturb key information in the hidden state, negatively impacting adaptation efficiency. Additionally, experiments with higher blur intensity ($\times 4$ Down-sampling) showed that excessive blur similarly degraded original image information, adversely affecting adaptation efficiency. Finally, we tested the effect of calculating pixel loss only in non-blurred regions (W/o blur loss). Omitting blurred region loss reduced PSNR by 0.05 dB, suggesting that calculating loss in appropriately blurred degrees effectively improves network robustness to degradation. In other words, when the network can manage severe blur degradation effectively, handling mild blur becomes easier.

## D.4 ABLATION EXPERIMENTS ON DIFFERENT BLUR AUGMENTATION RATIO

We conducted ablation experiments on the ratio of blur augmentation. As shown in Table 7, domain adaptation performance is optimal with a blur augmentation ratio of 0.6. Insufficient ratio produces few of blur blocks, failing to adequately disrupt the network, while an excessive ratio overly degrades image information, introducing too much interference in the hidden state, thereby negatively impacting adaptation efficiency.

Table 6: Ablation experiments on random blur data augmentation.

| Method | PSNR | SSIM |
|---|---|---|
| Opaque masking | 31.01 | 0.8615 |
| ×4 Down-sampling | 31.33 | **0.8643** |
| W/o blur loss | 31.47 | **0.8643** |
| DaMamba | **31.52** | 0.8642 |

Table 7: Ablation experiments on different blur augmentation ratio $Ratio^B$

| Ratio | PSNR | SSIM |
|---|---|---|
| 0.1 | 31.37 | **0.8653** |
| 0.2 | 31.39 | 0.8651 |
| 0.3 | 31.44 | 0.8650 |
| 0.4 | 31.47 | 0.8652 |
| 0.5 | 31.49 | 0.8644 |
| 0.6 | **31.52** | 0.8642 |
| 0.7 | 31.48 | 0.8648 |
| 0.8 | 31.45 | 0.8631 |
| 0.9 | 31.28 | 0.8610 |

## D.5 ABLATION EXPERIMENTS ON DIFFERENT RATIO OF SALIENT AND ORDINARY LAYERS

We conducted ablation experiments on the allocation ratio of salient and ordinary layers. As shown in Table 8, performance is optimal at a 0.6 ratio, meaning the proportion of salient to ordinary layers is 6:4. Insufficient salient layers results in many network layers having a low learning rate, leading to underfitting to the target domain. Conversely, an excessive ratio gives many layers a high learning rate, diminishing feature processing capabilities acquired during pre-training.

## D.6 ABLATION EXPERIMENTS ON TRAINING STRATEGY

We conducted ablation experiments on the training strategy. As shown in the Table 9, after removing the EMA strategy (w/o EMA), the PSNR decreased by 0.12 dB. The EMA strategy stabilizes network weight updates and mitigates the negative impact of outlier samples on the network. Additionally, removing the warm-up strategy led to a 0.19 dB decrease in PSNR (w/o Warm-up). Due to the introduction of the uninitialized SA module in the teacher network, directly using the teacher network to guide the target domain network results in incorrect pseudo-labels, leading to guidance deviation in domain adaptation. DAMamba divides the domain adaptation training process into a warm-up stage and a formal fine-tuning stage. In the warm-up strategy, pseudo-labels generated by

Table 8: Ablation experiments on different ratio of salient and ordinary layers. The adaptive learning strategy is used in both the warm-up and formal fine-tuning stages. Testing all possible ratios would require significant computational resources, so we conducted ablation experiments with different ratios only in the formal fine-tuning stage, setting the interval to 0.2 and adjusting it to 0.1 near the optimal ratio.

| Ratio | PSNR | SSIM |
|---|---|---|
| 0.1 | 31.42 | **0.8652** |
| 0.3 | 31.46 | 0.8650 |
| 0.4 | 31.48 | 0.8648 |
| 0.5 | 31.50 | 0.8645 |
| 0.6 | **31.52** | 0.8642 |
| 0.7 | 31.50 | 0.8642 |
| 0.9 | 31.49 | 0.8647 |

Table 9: Ablation experiments on training strategy.

| Ratio | PSNR | SSIM |
|---|---|---|
| w/o EMA | 31.40 | 0.8640 |
| w/o Warm-up | 31.33 | 0.8621 |
| DAMamba | **31.52** | **0.8642** |

Table 10: Robustness experiments on different network architectures.

| Method | PSNR | SSIM |
|---|---|---|
| MMA (Cheng et al., 2024) | 29.23 | 0.8309 |
| MMA + DAMamba | 31.14 | 0.8605 |
| MambaIR (Guo et al., 2025) | 30.83 | 0.8610 |
| MambaIR + DAMamba | 31.52 | 0.8642 |
| MambaCSR (Ren et al., 2024) | 31.24 | 0.8633 |
| MambaCSR + DAMamba | 31.49 | 0.8637 |

the source domain network are used to initialize the teacher network. Subsequently, in the formal fine-tuning stage, the initialized teacher network is employed to guide the target domain network, effectively preventing this deviation.

### D.7    ABLATION EXPERIMENTS ON DIFFERENT NETWORK ARCHITECTURES

We conducted domain adaptation training on various Mamba-based SR networks to validate the robustness of DAMamba across different architectures. As shown in Table 10, DAMamba achieved strong performance on MambaIR (Zou et al., 2024), MMA (Xiao et al., 2024a) and MambaCSR (Zhang et al., 2024b) networks. On the MMA network, domain adaptation training with DAMamba improved the PSNR by 1.91 dB compared to the source domain model without adaptation training.

### D.8    ABLATION EXPERIMENTS ON DIFFERENT DATASETS

We conducted domain adaptation training on various source and target domain datasets to validate the robustness of DAMamba across different data distributions. As shown in Table 11 and Table 12, DAMamba demonstrated strong robustness across various data distributions, with varying degrees of performance improvement.

### D.9    ABLATION EXPERIMENTS ON TRAINING/INFERENCE EFFICIENCY

As shown in Table 13, we comprehensively evaluated the baseline model (MambaIR), DAMamba, and the modified network (with the SA module replaced by a global attention module) in terms of computational complexity, parameter count, inference time per image, and training time per image. Due to the low computational complexity (linear) of SSM, the introduction of the SA module does not lead to a significant increase in computational cost. However, when the SA module is replaced with a self-attention module, which exhibits quadratic complexity with respect to the input sequence length, the computational cost increases substantially.

## E    VISUALIZATION RESULTS

As shown in Figure 9 and Figure 10, we compared the proposed DAMamba with other domain adaptation methods. DAMamba demonstrated superior detail restoration and noise suppression.

Table 11: Robustness experiments on different source and target domain datasets. → represents the domain adaptation from the source domain to the target domain.

| Method | PSNR | SSIM |
|---|---|---|
| Panasonic → Sony | 31.40 | 0.8854 |
| + DAMamba | 31.88 | 0.8867 |
| Panasonic → Olympus | 30.45 | 0.8576 |
| + DAMamba | 30.73 | 0.8595 |
| Olympus → Sony | 30.52 | 0.8654 |
| + DAMamba | 31.68 | 0.8801 |
| Olympus → Panasonic | 30.83 | 0.8610 |
| + DAMamba | 31.52 | 0.8642 |
| Sony → Panasonic | 30.61 | 0.8563 |
| + DAMamba | 31.36 | 0.8623 |
| Sony → Olympus | 30.50 | 0.8493 |
| + DAMamba | 30.72 | 0.8529 |

Table 12: Robustness experiments on different target domain datasets. The source domain corresponds to the Olympus camera branch.

| | B100 | | Mamga109 | | Set14 | |
|---|---|---|---|---|---|---|
| Methods | PSNR↑ | SSIM↑ | PSNR↑ | SSIM↑ | PSNR↑ | SSIM↑ |
| Baseline | 23.6560 | 0.6565 | 22.7882 | 0.8067 | 23.1563 | 0.6769 |
| DAmamba | 25.4150 | 0.6948 | 25.5669 | 0.8515 | 25.0627 | 0.7182 |
| | DIV2K | | Urban100 | | Set5 | |
| Methods | PSNR↑ | SSIM↑ | PSNR↑ | SSIM↑ | PSNR↑ | SSIM↑ |
| Baseline | 21.5328 | 0.6052 | 21.2945 | 0.6690 | 24.4840 | 0.7645 |
| MC-TTDG | 23.1358 | 0.6336 | 23.2613 | 0.7227 | 27.8980 | 0.8324 |

Table 13: Ablation experiments on training/inference efficiency

| Network | GFlops | Params (M) | Inf time (ms) | Training time (s) |
|---|---|---|---|---|
| Baseline | 112.99 | 20.56 | 63.61 | 1.4020 |
| Self-attention | 165.55 | 21.61 | 75.34 | 1.4775 |
| DAMamba (Ours) | 113.65 | 20.91 | 64.25 | 1.4173 |

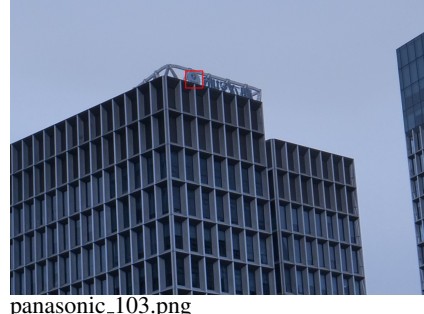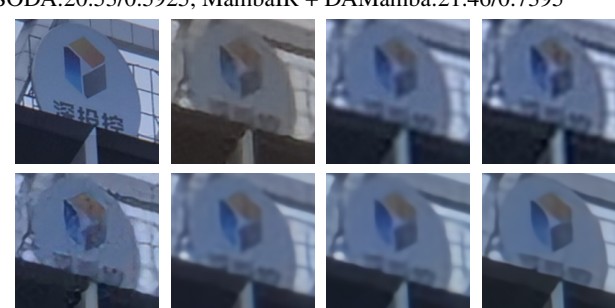

panasonic_187.png:
GT: None, CinCGAN: 20.58/0.5295, ZSSR:19.45/0.4983, SRTTA:18.96/0.4806
DASR: 19.05/0.5061, IODA:21.10/0.6584, SODA:20.53/0.5925, MambaIR + DAMamba:21.46/0.7395

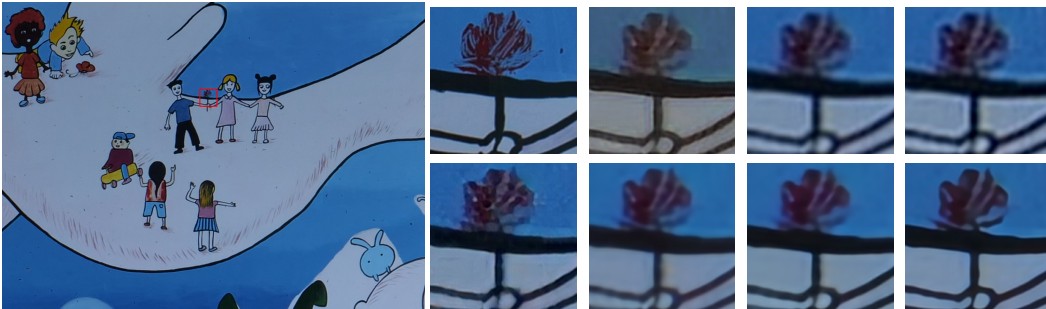

panasonic_103.png
GT: None, CinCGAN:26.88/0.8025, ZSSR:27.69/0.7863, SRTTA:27.38/0.7753
DASR:27.94/0.7898, IODA:28.27/0.8196, SODA:28.41/0.8264, MambaIR + DAMamba:28.85/0.8478

panasonic_128.png
GT: None, CinCGAN:27.15/0.7997, ZSSR:26.76/0.7512, SRTTA:26.46/0.7393
DASR:28.00/0.8028, IODA:27.04/0.8127, SODA:28.46/0.83, MambaIR + DAMamba:29.41/0.8537

Figure 9: Part 1 of the visualization display diagram. The large image on the left is the LR image, and the sub-images on the right are GT, ZSSR (Shocher et al., 2018), SRTTA (Deng et al., 2023)(first row), DASR(Wei et al., 2021), IODA(Tang & Yang, 2024), SODA (Ai et al., 2024), MambaIR (Guo et al., 2025) + DAMamba (second row). The values following the name represent the PSNR and SSIM metrics of the current patch. Please zoom-in on screen.

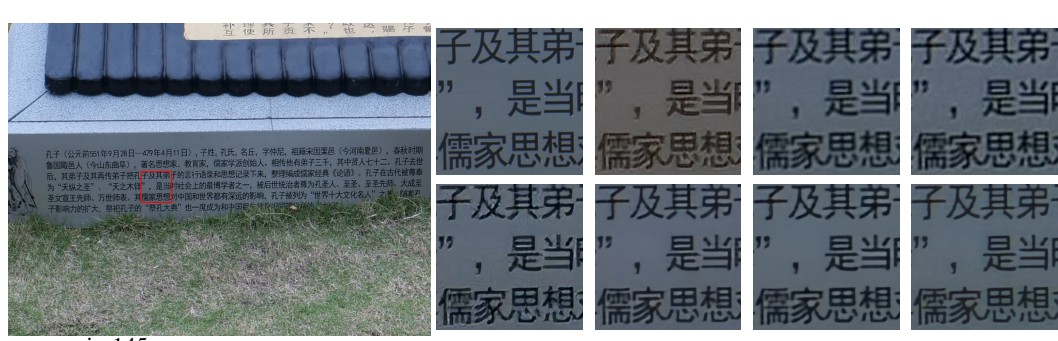

panasonic_145.png
GT: None, CinCGAN:28.58/0.8362, ZSSR:28.53/0.8402, SRTTA:27.80/0.8377
DASR:27.86/0.8437, IODA:31.62/0.926, SODA:30.61/0.9189, MambaIR + DAMamba:32.68/0.9388

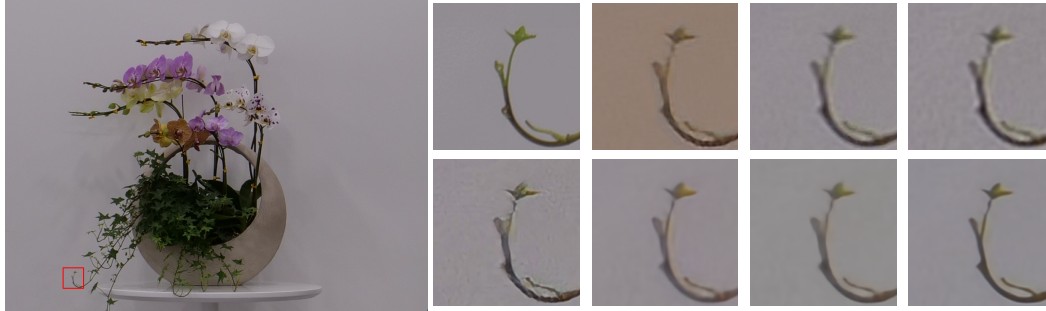

panasonic_158.png
GT: None, CinCGAN:27.01/0.9516, ZSSR:35.32/0.9467, SRTTA:34.78/0.9434
DASR:33.99/0.931, IODA:34.98/0.9619, SODA:35.59/0.9608, MambaIR + DAMamba:37.80/0.9723

Figure 10: Part 2 of the visualization display diagram. The large image on the left is the LR image, and the sub-images on the right are GT, ZSSR (Shocher et al., 2018), SRTTA (Deng et al., 2023) (first row), DASR(Wei et al., 2021), IODA(Tang & Yang, 2024), SODA (Ai et al., 2024), MambaIR (Guo et al., 2025) + DAMamba (second row). The values following the name represent the PSNR and SSIM metrics of the current patch. Please zoom-in on screen.

## F  STATEMENT ON LLM USAGE

We used a Large Language Model (LLM), specifically ChatGPT, solely for language polishing and improving the readability of the manuscript. The LLM was not used to generate ideas, conduct experiments, analyze results, or contribute to the research methodology. All scientific content, including the conceptualization, design, implementation, and validation of the work, was entirely carried out by the authors.

