# OpenReview forum: "DAMamba: Semantic Aware One-shot Test-time Domain Adaptation for Super-resolution"
_ICLR.cc/2026/Conference — ICLR 2026 Conference Withdrawn Submission_

### Official Review · Reviewer_rKHJ · 2025-10-29

**Soundness:** 3
**Presentation:** 3
**Contribution:** 3
**Rating:** 6
**Confidence:** 3

**Summary:**

This paper proposes DAMamba, a semantic-aware, one-shot, test-time domain adaptation method for Mamba-based super-resolution (SR) networks, designed to mitigate domain gaps when source data is unavailable and only a single target image is provided. The method employs a semantic prior-guided cross-training strategy using Alpha-CLIP to enhance key feature extraction and utilizes pairwise patch combinations to overcome the limited sample diversity of the one-shot setting. Additionally, it introduces a random blur data augmentation technique to improve the Mamba network's robustness and an adaptive learning strategy that dynamically identifies layer importance to boost adaptation efficiency. The authors claim that extensive experiments show DAMamba outperforms existing source-free domain adaptation methods, even when those methods use multiple target domain samples.

**Strengths:**

1.According to extensive experiments, DAMamba's performance using only a single target domain sample surpasses that of state-of-the-art (SOTA) source-free domain adaptation methods that require multiple target domain samples.

2.The method is tailored for real-world scenarios, respecting data privacy (source-free) and user convenience (one-shot).

3.It uses semantic guidance and adaptive learning for efficiency, while the custom blur augmentation ensures robustness specifically for the Mamba architecture.

**Weaknesses:**

1.The method requires performing an adaptation training step (including "cross-training" and "pairwise patch combinations") after the user provides a single image, which likely introduces significant computational delay at inference time.

3.The method relies on an external model (Alpha-CLIP) for semantic guidance, making it more complex. Furthermore, some components are specifically designed for the "unique" properties of the Mamba network, which may limit its transferability to other SR architectures.

**Questions:**

Please refer to the weaknesses.

---

### Official Review · Reviewer_mSMh · 2025-10-31

**Soundness:** 2
**Presentation:** 2
**Contribution:** 2
**Rating:** 2
**Confidence:** 5

**Summary:**

This paper addresses the highly challenging and practical problem of one-shot, source-free test-time domain adaptation (TTA) for Mamba-based super-resolution (SR) networks. The authors tackle the realistic scenario where, due to privacy and practical constraints, a pre-trained model must adapt to a new domain using only a single target image, without access to any source data. To this end, the paper proposes DAMamba, a TTA framework specifically designed for Mamba-SR.

**Strengths:**

1. The paper is well-written and logically structured. Despite the method's inherent complexity, the authors effectively articulate the entire workflow using clear diagrams (e.g., Figure 2) and a well-organized presentation.
2. The inclusion of pseudo-code for the main algorithms (Algorithms 1 & 2) is a significant strength that greatly enhances the method's reproducibility.
3. The framework is built on well-motivated and solid technical ideas. Leveraging Alpha-CLIP to provide global semantic guidance for the SSM's linear scan is a particularly clever solution. Furthermore, the adaptive learning rate and two-stage training strategy contribute to a stable and effective adaptation process.

**Weaknesses:**

1. The term "test-time" may be slightly misleading. While the Mamba architecture is efficient, the TTA process itself requires multiple training iterations. The reported training time (~1.4s per image in Table 13) introduces a non-negligible latency for applications requiring an immediate response. "One-Shot Fast Fine-Tuning" might be a more accurate description of the process.
2. The core idea of the Semantic-Aware (SA) module is to prepend global semantic information before the SSM scan. While this is a simple and effective technique, its novelty lies more in its application to this specific problem rather than in a new architectural design. The contribution could be framed more precisely.

**Questions:**

1. The ablation study in Table 1 effectively validates the contribution of each component individually. Have you investigated any synergistic effects between them? For example, is the adaptive learning rate strategy most effective when specifically combined with the semantic-guided cross-training (SPCT)?

2. Many of the proposed ideas, such as using Alpha-CLIP for guidance and the pairwise patch combination strategy, seem broadly applicable. Could you comment on the feasibility and potential challenges of adapting the DAMamba framework to non-Mamba architectures, such as those based on Transformers or CNNs?

---

### Official Review · Reviewer_yneV · 2025-10-31

**Soundness:** 3
**Presentation:** 2
**Contribution:** 2
**Rating:** 4
**Confidence:** 5

**Summary:**

This paper introduces DAMamba, a semantic-aware, one-shot, test-time domain adaptation method for super-resolution tasks based on Mamba architectures. The approach leverages Alpha-CLIP-derived global semantic priors to guide the adaptation of Mamba-based super-resolution networks in the challenging setting where only a single target domain image is provided and source samples are inaccessible. Core contributions include a semantic-prior-guided cross-training strategy, a random blur augmentation addressing Mamba's contextual dependencies, and an adaptive learning protocol differentiating salient from ordinary layers for efficient domain adaptation. Extensive experiments on real-world datasets, comprehensive ablations, and qualitative analyses support the claims.

**Strengths:**

1. Practical and meaningful task setting. The paper tackles an important and realistic challenge: one-shot domain adaptation for test-time super-resolution when only a single (unlabeled) target domain image is available and no access to source data is assumed. This formulation is highly relevant for privacy-sensitive and user-centric applications.

2. Thoughtful methodological design. The composite method—pairwise patch cross-training for diversity, random blur augmentation tuned for Mamba's recurrent hidden dynamics, and adaptive learning of salient layers—reflects a deep understanding of both the problem and model class.

3. Comprehensive ablation study. The authors carefully tease apart each component of there method, e.g., effect of different semantic prior methods, SA module position, blur ratios, and training strategies. The empirical evidence is robust.

**Weaknesses:**

1. Lack of deeper theoretical analysis. While mathematical operations and implementation are made explicit, the paper stops short of providing any rigorous analysis of why the semantic prior-guided cross-training strategy provably prevents mode collapse or guarantees generalization under single-sample adaptation. The empirical studies are convincing, but deeper theoretical support is lacking.

2. Unclear justification and robustness of semantic guidance. The paper's rationale for using high-level semantic information (from CLIP) to guide a low-level pixel reconstruction (SR) task is underdeveloped. A conceptual gap exists: it is unclear how high-level semantics (e.g., "this is a building") can precisely guide low-level texture reconstruction (e.g., "the window edges should be sharper"). The method heavily relies on the quality and relevance of Alpha-CLIP's semantic priors. If the semantic prior extractor is itself sub-optimal or misaligned to the new domain, DAMamba could fail hard, and there is limited exploration of this failure mode. Furthermore, the authors' explanations for related concepts, such as "low-level semantics" (line 156), are confusing and require further clarification.

3. The main performance comparisons and ablation studies in the main text are conducted almost entirely on the DRealSR dataset, with SOTA comparisons notably limited to the specific  Olympus (source) to Panasonic (target) combination. It is unclear if this specific cross-brand shift (Olympus $\rightarrow$ Panasonic) is representative of all real-world domain gaps, which might be relatively limited. The method's effectiveness remains unproven for more significant shifts, such as adapting from a high-quality synthetic source (e.g., DIV2K + BSRGAN) to a real-world dataset.

4. Some design choices insufficiently justified. Decisions such as the architecture choice and positioning of the SA module (Section 2.3.3, Table 5), the 0.6 blur augmentation ratio (Table 7), and the salient-to-ordinary layer ratio (Table 8) are derived from ablation, but the rationale is empirical and a more systematic or generalizable principle might strengthen the work. Why these specific ratios? Are they robust across very different domains or model sizes?

5. In the appendix Table 12, the paper presents experiments on standard SR benchmarks (Set5, Set14, B100, Urban100). This evaluation is questionable. The core premise of the paper is domain adaptation —addressing the degradation distribution shift between training and testing data. However, Set5, Set14, etc., are "clean" datasets, and the domain gap between them and the DRealSR (Olympus) source domain is not a meaningful scenario for evaluating adaptation.

6. Discussion and positioning could be more forward-looking. The Related Work section (Appendix A) is exhaustive, but the main text spends little time critically positioning DAMamba against the fast-evolving landscape of zero-shot, semantic-driven, or multi-modal domain adaptation. More discussion of where DAMamba's limitations are likely to be most severe (e.g., out-of-distribution semantic priors, non-visual domains, non-image SR) would improve clarity.

**Questions:**

See weakness

---

### Official Review · Reviewer_qX7j · 2025-11-01

**Soundness:** 2
**Presentation:** 2
**Contribution:** 2
**Rating:** 4
**Confidence:** 3

**Summary:**

In order to address the problem of prohibited access to source domain samples and tendency of users to capture or upload only a single image for target domain SR, this paper focuses on test-time domain adaptation methods and employs semantic priors to increase SR performance. Extensive experiments demonstrate the effectiveness of this work, surpassing multi-sample SOTA source-free domain adaptation methods.

**Strengths:**

1. Clear Motivation: The motivation behind the proposed method is clearly articulated.
2. Extensive analyzing experiments: Present evaluations of the proposed framework such as ablation experiments of different strategies and comparison with SOTA methods.

**Weaknesses:**

1. Insufficient methodological clarification：
Regarding the issue of limited feature diversity, the paper introduces auxiliary patches to assist primary patches. However, the rationale behind how this strategy improves feature diversity is not clearly analyzed. Moreover, the paper lacks experimental comparisons demonstrating the performance difference between using both primary and auxiliary patches versus using only primary patches.

2. Analysis of experimental results requires improvement：
The visual comparisons provided lack convincing evidence. Specifically, the visual results of the baseline method (i.e., MambaIR) are missing in Appendix E, making it difficult to assess the actual improvement achieved by the proposed approach. Furthermore, the presented visual results do not exhibit clear qualitative advantages over other methods. It is recommended to include more representative examples that show noticeably sharper textures or more distinct details to strengthen the visual evidence.


3. Writing issues:
1) In line 079, “We”should be changed to“we.”
2) In Figure 2, ensure consistent capitalization — e.g., “Geometric augmentation,” “Random crop.”
3) Improve typesetting consistency: fix the caption formatting in Figure 6 and align the numerical values properly in Table 3.

**Questions:**

Please refer to the weakness part.

---

### Note · Authors · 2025-11-26

I have read and agree with the venue's withdrawal policy on behalf of myself and my co-authors.